



# Warming Climate Shortens Ice Durations and Alters Freeze and Breakup Patterns in Swedish Water Bodies

Sofia Hallerbäck[1,2], Laurie S. Huning[3,1], Charlotte Love[1], Magnus Persson[2], Katarina Stensen[4], David Gustafsson[4], Amir AghaKouchak[1,5]

[1] Department of Civil and Environmental Engineering, University of California, Irvine, USA

[2] Lund University, Sweden

[3] Department of Civil Engineering and Construction Engineering Management, California State University, Long Beach, USA

[4] Swedish Meteorological and Hydrological Institute, Sweden

[5] Department of Earth System Science, University of California, Irvine, USA

*Correspondence to*: Amir AghaKouchak (amir.a@uci.edu)

**Abstract.** Increasing air temperatures reduce the duration of ice cover on lakes and rivers, altering their water quality, ecology, biodiversity, and physical, economical and recreational function. Using a unique in-situ record of freeze and breakup dates, including records dating back to the beginning of the 18th century, we analyze changes in ice duration (i.e.,
first freeze to last breakup), freeze and breakup patterns across Sweden. Results indicate a significant trend in shorter ice duration (62%), later freeze (36%) and earlier breakup (58%) dates from 1913-2014. In the latter 3 decades, the mean observed ice durations have decreased by about 11 days in northern (above 60N) and 28 days in southern Sweden, whereas the average freeze date occurred about 10 days later and breakup date about 17 days earlier in southern Sweden. The rate of change is roughly twice as large in southern Sweden as in its northern part. Sweden has experienced an increase in
occurrence of years with an extremely short ice cover duration (i.e., less than 50 days), which occurred about eight times more often in southern Sweden than previously observed. Our analysis indicates that even a 1 °C increase in air temperatures in southern (northern) Sweden results in a mean decrease of ice duration of 22.5 (7.6) days. Given that warming is expected to continue across Sweden during the 21st century, we expect increasingly significant impacts on ice cover duration and hence, ecology, water quality, transportation, and recreational activities in the region.



## 1 Introduction


The world's freshwater systems are critically important for all humans and they have experienced significant environmental changes due to human activities and anthropogenic climate change (Dudgeon et al. 2006) (Vörösmarty et al. 2010). In the artic regions climate change is amplified, mainly due to temperature feedbacks and change in surface albedo (Pithan et al. 2014). A global mean temperature increase of +2 °C from the 'pre-industrial' level would result in a higher mean

temperature increase in Scandinavia (Vautard et al. 2014). Climate change therefore poses increasing stress on our freshwater resources. As temperatures are projected to continue to rise, they are expected to cause major physical, ecological, social, and economic changes (Parmesan and Yohe 2003).

One impact of global warming is the reduction in ice cover duration of freshwater systems, often associated with both a later freeze and an earlier breakup (Duguay et al. 2003). Since the ice duration of inland waters is strongly correlated with and

driven by air temperature and solar radiation (Sharma et al., 2019), (Kropácek et al. 2012), (Duguay et al. 2015), (Livingstonea and Adrianb 2009), the duration of ice cover on lakes and rivers is sensitive to climatic change and variability (Prowse et al. 2011), (Benson et al. 2012), (Beltaos and Prowse 2009), (Takács 2011), (Latifovic and Pouliot 2007), (Magnuson et al. 2000). Decreased ice cover duration and earlier ice breakup have critical ecological consequences, influencing the timing of photosynthesis (Quayle et al. 2002) (Leppäranta et al. 2012), productivity and biodiversity of

phytoplankton, and the occurrence of fish kill (Leppäranta et al. 2003), (Watz et al. 2016), as well as shaping the vegetation of flowing waters (Lind et al. 2014). Changes in ice duration also modulate the energy and moisture exchange over the water surface (Duguay et al. 2003) and the timing of vertical mixing and stratification in lakes (Bengtsson 2011).

In Sweden, lakes cover about 9% of the total land area (Henestål et al. 2015) and surface water provides 50% of its drinking water (Rosborg 2015). In many high-latitude regions, including parts of Sweden, ice cover is also an important part of

transportation and communication (Jeffries et al. 2012, Knoll et al. 2019). The Swedish Transport Administration, Trafikverket, has expressed concerns regarding the stability and duration of ice roads (i.e., frozen water bodies) and has requested that further research be conducted on this topic. The mean temperature of winter months in Sweden are expect to increase more than the summer month, with models predicting that a global mean temperature increase of +2°C from pre-industrial would results in an increase of +1.5 in the south to +3 °C in the north of Sweden compared to mean temperature of

1971-2000 (Vautard et al. 2014). Thus, reliable information about the rate of change in ice cover duration, which we investigate here, is needed to advise policy and decision makers, inform impact and adaptation activities, and understand potential impacts on recreational activities (e.g., ice fishing, ice boating, skiing, spiritual ceremonies, and ice skating) (Kling et al. 2003), (Knoll et al. 2019).

Previous studies have examined ice covered water bodies in Sweden. For instance, Eklund (1999) provided an overview of

ice data, up to 1999, and described methods of ice cover observation. Weyhenmeyer et al. (2004) also analyzed ice-breakup data across Sweden and identified regional differences in the timing and length of ice cover across the country's water



bodies. They concluded that cooler regions (i.e., northern Sweden) are less sensitive to increasing air temperature than temperate regions (i.e., southern Sweden) in terms of earlier breakup of ice cover. Comparing the colder time period of 1961–1990 to the warmer period of 1991-2002 (by 0.8°C), they observed that ice breakup occurred 17 days earlier in the

southern region during the latter period, but only 4 days earlier in the northern region. Hence, they concluded that increasing air temperature can drastically shift the timing of lake ice breakup in the warmer and southern regions of Sweden, while it is less drastic in the northern parts. Since a long time series of lake ice observations can serve as a proxy for climate change and provide a convenient climate index (Duguay et al. 2003), we extend these previous studies using more recent observations, longer records of ice duration, breakup, and freeze data, and more observation sites across the whole of

Sweden.

Weyhenmeyer et al. (2011) investigated the ice phenology from 1213 lakes and 236 rivers in 12 different countries, including Sweden. They find a stronger decrease in ice duration across regions with shorter durations and air temperatures below 0 ℃, such as southern Sweden below 61°N. They suggested that 3.7% of the world's lakes larger than 0.1km$^2$ are at risk of becoming open-water systems in the near future. Sharma et al. (2019) estimate that the number of lakes with

intermittent winter ice cover are increasing, from currently 14,800 lakes in the Northern Hemisphere to 35,300 with 2 °C warming and 230,400 with 8 °C, impacting up to about 400 and 650 million people.

In this study, we characterize the ice cover duration using in situ observations, from the Swedish Meteorological and Hydrological Institute, of ice cover freeze and breakup from 752 lake and river observation sites in Sweden (see the Methods and Data Section) to better understand the relationship between temperature variability and ice duration and associated

trends. We quantify temporal and regional shifts in the distributions of ice freeze and breakup dates and ice durations across Sweden since they remain poorly understood. The ground-based observations we use consistently include the first day of complete ice cover and/or the last day of complete breakup. Following Eklund (1999), we define the "freeze date" as the first date from which the whole lake, or the whole part of the lake, except for small segments, is ice covered for a minimum of three consecutive days. Moreover, we define the "breakup date" as the last date, following complete ice cover, that the entire

lake becomes free of ice, with the exception of small parts of ice close to the shore and around obstacles (Eklund 1999). In this study, we define ice duration as the number of days between the first freeze date and the last breakup date.

The observations include lakes and some rives with a variety of size. As for big lakes, such as Vänern (5,650 km²) or Mälaren (1,072 km²), a smaller somewhat separated part, often a bay, is observed. Although it is not uncommon for multiple ice cover periods to occur in one year, especially in southern Sweden, the definition of ice duration in this study does not

take this into account. Moreover, we do not separate a year with no ice cover from a year with no data, since no records exist for years without ice cover. The record length of the freeze and breakup data varies across the observation sites, with observations spanning from 1700-2014.



## 2 Methods and Data

### 2.1 Ice Data Observations, Monitoring, and Uncertainty

We used ice data records obtained mainly from the Swedish Meteorological and Hydrological Institute (SMHI), which include ice freeze and/or breakup dates. Torne river observations are made by Finnish Environmental Institute SYKE. In total, the dataset includes observations from 752 observations sites, consisting of mainly lakes and some rivers. Observations span from 1700-2014, however the coverage varies over time and is a factor of uncertainty. The number of observations from 1860-2014 are presented in Figure 3.

Since 1870 lake ice was observed systematically and manually from the shore by an observer responsible for monitoring that lake or the part or bay of the large lake (Eklund, 1999). Some records date back even longer and come for example from records taken from boats. Even though some water bodies may freeze and melt several times over a season, the data record we used only includes two dates per season.

For years with multiple observations (occurring in digitalized data from 2012-2014), the first observation of freeze and the
last observation of breakup were chosen to be consistent with the previous years of data. Some larger lakes have more than one observation site, however we evaluated the sites separately. Years with no observations were excluded from analysis and years with no ice were not reported in the data.

The ice year in the study is defined from August 31st, were September 1st is the first day of the ice year. The date was selected for practical reasons after finding a minimal number of observations of freeze dates prior to August 31st or breakup
dates later than August 31st. Ice duration was derived as the number of days between the freeze and breakup date, for years and sites with both a freeze date and breakup date observation.

According to SMHI, the potential uncertainties in the ice data are associated with the manual observations, different observers over time, and digitalization. SMHI are aware that the definition of freeze and breakup during digitalization of the data might not always be consistent. During the period 1980-2000, the longest consistent ice-period occurred for some sites
during the period of record digitalization with multiple observations, which as noted above, could impact our results. From year 2000 onward, the digitalization is again consistent with the original definition of first freeze and last breakup.

### 2.2 Statistical Tests, Trends, and Spatial Analyses

Records were tested for statistically significant trends over 1913 to 2014 for ice duration, freeze and/or breakup dates. A significance level of 95% was used to evaluate trends in the ice phenology variables (freeze date, breakup date, and ice
duration). The Mann-Kendall R package 'Kendall' (McLeod 2011) was used to perform the Mann-Kendall trend tests (Kendall 1938). Observation points with at least 81 years of recorded data from the 101-year time period (maximum of 20% missing data) were used in the analysis.





We analysed the mean breakup dates over ten-year intervals from 1871 to 2010 (Figure 1) using all observations over the time period and implemented bivariate interpolation onto a grid for irregularly spaced input data. Bilinear interpolation is
applied using algorithms from the R package 'Akima' and 'interp' function. The mean value was used for observations sites with more than one year of data (duplicates). As a result, each observation site is only represented once; however, the mean value used can be a combination of one to ten years of data for that point.

## 3 Results

### 3.1 Earlier ice breakup since the end of the 19th century

We analyze all breakup date observations from 1871 onward, since systematic observations started in 1870 (Eklund, 1999). Figure 1 presents the mean breakup dates from 1871 to 2010, which complements the changes and trends in ice cover described above. We generated the maps shown in Figure 1 by bi-linearly interpolating the data gathered by the Swedish Meteorological and Hydrological Institute (SMHI) across the country. As time progresses in Figure 1, the area with breakup dates after June decreases (blue contour), and the areas of breakup prior to April (purple contour) and May (red contour)
move northward. From 1871 to 1930 and 1951 to 1970 the geographical mean breakup pattern is similar, while the years 1931 to 1950 display a larger area with breakup before April. Earlier breakup even extends into the Arctic circle. The largest difference in breakup dates are found in the later period of 1991 to 2010. In the last time period 2001-2010 the ice breakup generally occurred prior to the first day of June even in the northers part of Sweden.

We consider a river with observations dating back to the 18[th] century, the Torne River, located in northern Sweden bordering
Finland (Johansson 1932, Kajander 1995, Korhonen 2006). The recorded 291 years of breakup dates and 37 years of freeze dates were recorded for the lower portion of the Torne River close to Haparanda. Observations of ice breakup are used from 1700 to 2009, however, observations were unavailable from 1770-1789. We find a significant trend in earlier ice breakup using the Mann-Kendall trend test with a 99% significance level ($p \ll 0.01$). Sharma et al. (2016) showed that the Torne river in northern Sweden and Finland has experienced earlier breakup and an increase in abnormally early breakup dates over the
last 320 years.

In addition, we consider the Västerås fjärd, which is a waterway in the northwest part of the brackish lake Mälaren close to the city of Västerås just below 60 $^{\circ}$N in Sweden. Despite 38 years of missing data, with the longest consecutive period of missing data being the 10-year period from 1974-1985, the Västerås fjärd has long-term records with 234 years of breakup data (1711-2012) and 69 years of freeze dates (1870-1986). The lake displays a significant trend in later freeze (p-value =
0.002); however, we do not find a significant trend in earlier breakup. See Figure S1 in the supplementary material for breakup observations from Västerås fjärd and Torne River.





Further, we analyze all water bodies with observations of freeze and subsequent breakup dates in Sweden. For each day of the ice year (September-August) from 1860-2014, Figure 2 shows the fraction of observed water bodies across Sweden that are ice covered. In other words, the fraction represents the total number of ice observations for that specific day divided by

the total number of observations available that year. To better understand the relative fraction of ice covered water bodies for a given year, which ranges from zero (days with no observed ice cover) to unity (days where all observed sites were ice covered), Figure 2d quantifies the number of available observations used per year. Figure 2 further demonstrates that a decrease in ice cover duration has occurred since the 1860s. Moreover, since the late 1980s, there is a noticeable increase in inter-annual variability and years with extremely short ice cover duration, especially in southern Sweden (see Figure 2),

which we further investigate below.

### 3.2 Significantly reduced ice duration

Now we characterize the water bodies with minimal missing data by analyzing trends in the observed ice duration and freeze and breakup dates (i.e., ice phenology variables) from 1913-2014 using the Mann-Kendall trend test (Kendall 1938) at a 95% significance level, Figure 3. Fifty-eight water bodies with a maximum of 20% missing data during the 101-year period are

included. Since the Mann-Kendall trend test is sensitive to missing data, and using a dataset with missing data is a source of uncertainty, we excluded many observations sites in this analysis to have higher confidence that the observed change represents a change over the given time period.

Forty water bodies displayed a significant trend in at least one of the ice variables associated with a warmer climate (i.e., decreasing ice cover duration (Figure 3a), later freeze (Figure 3b), or earlier breakup (Figure 3c). Fourteen of these water

bodies showed a significant trend in all three ice variables associated with a warmer climate. Out of these 14 sites, all except one, are located above $60^{\circ}$N (i.e., northern Sweden). A significantly decreasing trend in ice cover duration was found for 24 of the 40 sites (62.5% of sites). A significant trend in later freeze date was seen in 17 out of 47 (36%) sites and an earlier breakup was observed in 32 out of 55 (58%) sites.

Sites with a significant trend were located both in southern (latitudes $< 60^{\circ}$N) and northern Sweden, and included lakes of a

variety of sizes. The sites with longer records more commonly exhibited a trend than those with more missing data. We did not find any statistically significant increasing trends in ice duration, nor later breakup for any of the lakes or rivers. Hence, our findings indicate that the ice duration is shrinking over the majority of our study sites in Sweden with earlier ice breakup and later freeze occurring. Only one lake has a trend of earlier freeze, the lake Flåsjön in northern Sweden (blue mark in Figure 3b). The heavily regulated lake Flåsjön displayed a shift in timing of ice cover, with both a significant trend showing

an earlier freeze and an earlier breakup. Overall, according to SMHI harmonized temperature data the mean change in air temperature in Sweden was about +0.09 $^{\circ}$C per decade from 1901-2014.



Furthermore, we observe a rate of change in ice duration of -1.9 days per decade across Sweden from 1914-2014. For comparison, Table S1 (Supplementary Materials) summarizes the changes in ice duration, freeze date and breakup date from previous studies on inland ice cover; Magnusson et al. (2000), Benson et al. (2012), Takács (2001), Latifovic et al. (2007), Jenset et al. (2007) and Hodgkins (2013). Moreover, for our later study period, 1959-2014, the corresponding rates of change for later freeze and earlier breakup dates across Sweden are three and four times faster, respectively, than those Magnuson et al. (2000) found for lakes across the Northern Hemisphere from 1846-1995 (Table S1). Comparing our two time periods (i.e., the entire period of 1914-2014 with the last 56 years of 1959-2014), the rate of change has increased in Sweden, especially in southern Sweden where it is warmer (Table S1).

### 3.3 Changes in timing of ice cover

Figure 4 compares the distributions of observed ice duration and freeze and breakup dates for the two periods 1955-1984 and 1985-2014. The analysis includes all observed sites with a maximum of 20% missing data in the 30-year periods. Figures 4a and 4d demonstrate that the ice duration has decreased over the last 30 years, with a mean decrease of about 11 and 28 days in northern and southern Sweden, respectively. The shift in the mean ice duration is larger in southern Sweden compared to the northern region. Although shorter ice durations are more common in both regions during 1985-2014 than 1955-1985, the shape of the ice duration distribution between the two time periods also more greatly differs in southern Sweden. Over the more recent 30-year period in southern Sweden, the mean freeze date occurs 10 days later (Figure 4e), while the mean breakup date occurs 17 days earlier (Figure 4f) than previously observed. Considering the same time periods, the shifts in the mean timing of freeze and breakup are smaller in northern Sweden, with the mean freeze date observed about four days later (Figure 4b), and the mean breakup date occurring five days earlier (Figure 4c) in the last 30 years. Comparing the mean temperature of the two time periods using SMHI homogenized data, the mean temperature of the later period (1985-2014) is about 0.85 $^{o}$C warmer than during the earlier period.

Changes in the timing of ice cover have shifted the mean duration, resulting from an increase in extremely short durations (Figure 4). Figure 4d indicates that during 1985-2014 southern Sweden experienced a 750% increase in the number of years with less than 50 days of ice cover relative to the previous 30 years. In northern Sweden (Figure 4a), an increase of 300% was observed in the number of years with an ice cover duration of less than 100 days. Hence, Figure 4 highlights differences in the ice duration distributions related to the geographical locations of the observations (i.e., northern vs. southern Sweden), as well as the freeze date and the morphological aspects. Nonetheless, other lake characteristics such as mean depth, volume, and area also influence the duration of ice cover and its freeze and breakup dates.

An increase in the interannual variability of ice cover duration could be an early warning sign of ecological regime shift (Carpenter et al. 2011). Magnuson et al. (1997) demonstrated that such an increase in interannual variability and more frequent, extremely short durations of lake ice cover have occurred since the 1950s across Canada, the United States,





Finland, Switzerland, Russia, and Japan. We also find an increase in interannual variability across Swedish lakes and rivers. The observations in Figure 4 can be related to change in both mean and variance described by Benson et al. (2012).

**3.4 Change in ice duration per 1°C of warming**

To better understand the drivers of variability in the ice cover durations observed above, Figure 5 characterizes the relationship between the local mean annual air temperatures (September-August) and ice durations using about 30,000 ice duration and air temperature observations from the Climatic Research Unit (CRU) v 3.23 (Harris et al. 2014). These relationships are considered with respect to latitude (Figure 5a) (719 water bodies), as well as mean depth (Figure 5b),

volume (Figure 5c) and area (Figure 5d) (464 lakes) – see Appendix for details. Water bodies in southern Sweden experience warmer temperatures, and larger water bodies (volume, depth, or area) generally have shorter ice durations compared to smaller ones. Larger water volumes in lakes, for instance, might result in a later freeze date and therefore contribute to a shorter ice duration. Figure 5 also shows that water bodies experiencing higher mean annual air temperature have shorter ice durations.

We quantify the change in ice cover duration relative to air temperature for 464 lake water bodies using temperature and ice data from 1901-2014, Figure 6 – see Appendix for details. We applied linear regression to obtain the rate of change in ice duration for each lake relative to a $1°C$ increase in air temperature. We found a greater shift in ice cover duration per $1°C$ increase for lakes in southern Sweden than at higher latitudes. In southern Sweden, mean ice duration decreases by 22.5 days per $1°C$ increase, with the mean breakup date occurring 11.2 days earlier, and the mean freeze date occurring 8.1 days later.

In northern Sweden, the shift per $1°C$ increase is smaller, with mean values for ice cover duration decreasing by 7.6 days, the freeze date occurring 3.6 days later, and the breakup date occurring 4.4 days earlier per degree of warming.

**4 Discussion and Conclusion**

We show that for many water bodies in Sweden, freeze dates are occurring later in the season while breakup dates are occurring earlier (Figures 1-4). As a result, the ice cover duration has decreased in many of the observed lakes (Figures 2-4).

Moreover, we find an increase in the number of years with extremely short ice cover duration (Figure 4), especially in the last 30 years (i.e., 1985-2014) in southern Sweden (latitudes < $60°N$). Similar results are also found in Finland, with a larger change in southern Finland compared to northern (Korhonen et al. 2006). The breakup date is similar for lakes in similar geographic regions and is not dependent on the size of the lake (Eklund 1999). As shown in Figure 1, earlier ice breakup observed over all of Sweden.

Earlier ice breakup affects the productivity and biodiversity of phytoplankton (Leppäranta et al. 2003), reduces winter dependent zooplankton species (Kling et al. 2003), and decreases fish production while increasing winter fish deaths (Watz et al. 2015), (Leppäranta et al. 2003). Moreover, with a decrease in ice duration, the photosynthesis in the lakes and rivers



extends longer into the fall season and starts sooner in the spring (Kling et al. 2003). Longer durations without ice cover have a larger impact on the mixing patterns of the lake. Moreover, the depth of the mixing layer is related to light availably,
phytoplankton density, and the carbon to nutrient ratio of phytoplankton, as well as zooplankton biomass, which may have effects on higher trophic levels (Berger et al. 2006).

We demonstrate that ice cover duration has reduced in Sweden, and thereby the periods with suitable conditions for the use of frozen rivers and lakes as ice roads have decreased by approximately seven days of ice cover per degree Celsius of air temperature increase. Here, we investigate changes in the ice cover duration from first complete ice cover to complete
breakup; however, further investigation of changes in ice thickness would be beneficial for assessing the safety of transportation over ice as well as the occurrence of multiple ice periods. Similarly, while the season for ice-related recreational activities will shorten, again, other factors such as ice thickness should be accounted for the safety of individuals partaking in activities on frozen water bodies.

In Sweden, a more drastic shift in ice breakup dates is expected with increasing air temperature in temperate regions
(average air temperatures of 5 to 7°C) compared to colder regions (-2 to 2°C) (Weyhenmeyer et al. 2004). Weyhenmeyer et al. (2004) found that the nonlinear relationship between breakup date and the mean annual temperature across Sweden can be represented using an arc cosine function. They also analyzed the amplitude of annual air temperature cycles as a proxy for solar radiation, concluding that in climate regions with larger temperature differences between winter and summer months, the sensitivity of ice duration to temperature changes is smaller than in regions with a smaller annual air temperature
amplitude. Thus, lakes and rivers in the southern part of Sweden are more vulnerable to increasing temperature than regions with a larger temperature difference between summer and winter months. Our results reveal that drastic changes in southern Sweden appear to already be occurring (Figures 1-4).

During the period of 1959-2014, southern Sweden exhibited a large rate of change in ice duration relative to the studies found and referenced herein (Table S1 Supplementary Materials). This might be due to the vulnerability of lakes in regions
with a smaller temperature amplitude (Weyhenmeyer et al. 2004). In addition to the warming air temperatures found across Sweden (Moberg and Alexandersson 1997), (Alexandersson and Moberg 1997), many other cold regions across the world are experiencing, or are expected to experience, increases in air temperatures. Therefore, it is worthwhile to expand similar investigations to the seasonally frozen lakes and rivers in other cold regions to determine if they are also experiencing an accelerating rate of decrease in ice cover duration, since such rates may lead to significant changes extending beyond the
local biogeochemical processes of the water body, especially if observed globally.

Moreover, air temperatures at high-latitudes are increasing more rapidly than the global mean temperature, making studies related to lake and river ice increasingly important. Additionally, historical data and model predictions indicate that the average temperature of cold days has increased more than the mean temperature in eastern and northern Europe (Vautard et





al. 2014, Kjellström 2004). Hence, the winter climate of this region is likely to be affected more by climate change than the
annual mean temperature.

Future research is needed to assess the influence of reduced ice cover duration on the lake and river water quality and
ecology including potential eutrophication, acidification, regulation, and flow pattern changes. The consequences of the
release of nutrients and oxygen-demanding substances cannot yet be forecast for ice covered lakes (Bengtsson 2011). Given
the significant changes in ice cover duration that we characterized across Sweden, incorporating ice cover into the overall
picture of impacts on freshwater resources may be helpful for improving the understanding of variability within freshwater
systems.

Regulation of lakes and rivers may also influence the trend in freeze and breakup dates, potentially leading to either an
earlier or later freeze date than would naturally occur. Although, hydroelectric power plants expanded in Sweden in the
period of 1950-1970, regulation is typically constant over time once it is established.

**Appendix**

We use the air temperature data from the Climatic Research Unit (CRU) v 3.23 (Harris et al. 2014) to assess the relationship
between temperature and all of the freeze date, breakup date, and ice duration observations (Figure 5). The spatial resolution
of the CRU monthly mean air temperature data is 0.5x0.5 degrees and the temporal coverage is from 1901 to 2014. The
average air temperature for the ice year was derived using the average of the monthly mean values from September to
August.

We coupled the ice data record with information from SVAR (Swedish Water Archive 2012, SMHI) to derive information
about the waterbody (such as location, areal extent, depth etc.). The geographic location and shape of the lakes was used to
extract corresponding data from the lakes, using the shapefiles from SVAR (Swedish Water Archive 2012, SMHI). For
larger lakes overlapping several CRU grid cells, we used the average temperature value of those cells to represent the air
temperature over the lake (Figures 5 and 6). Our analysis includes all ice observations within the time period, where big
lakes with multiple observations and lakes with more observations are more frequently represented than others (Figures 5
and 6).

Moreover, we used ordinary least squares via the lm() function in the R package 'stats' to compute the slope of the line of
best fit for the temperature and ice phenology variables. The slopes serve as a measure of the change in the number of days
(of ice duration, freeze date, or breakup date) per 1 °C temperature increase (Figure 6). The analysis for different lakes varies
over observed years and temperature ranges. Since the latitude in Sweden roughly represents the climatic zones, the latitude
was included in Figure 6.



Ice duration was coupled with mean annual temperature from September to August. The mean correlation of ice duration and air temperature (September-August) was r = -0.54. Freeze date was coupled with mean temperatures from October to

December, and breakup dates with mean temperatures from March to May. The breakup date has a stronger mean correlation with temperature (r= -0.69, temperatures March-May) compared to the freeze date (r=+0.50, temperatures October-December).

**Acknowledgments**

This study was conducted at the University of California, Irvine, in collaboration the Swedish Meteorological and
Hydrological Institute and Lund University. The US authors were partially supported by the NOAA MAPP program. We especially want to thank all who have contributed to the collection and curation of in situ ice cover observations over the years. Also, we want to acknowledge previous work by Eklund (1999) which is the base of the big data set used in this analysis. The study was a part of the master thesis by SH; thank you to all who contributed with great questions, interest and discussions along the way at the University of California Irvine, SMHI and Lund University.

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

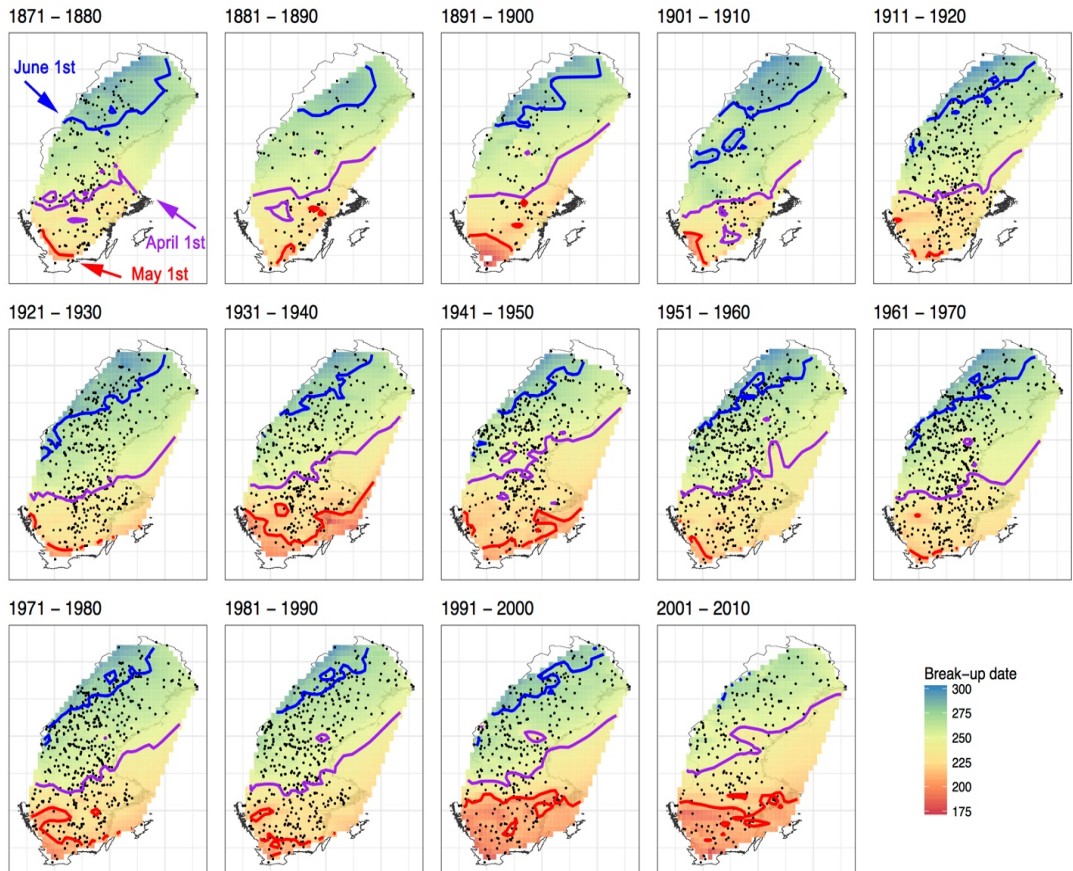

**Figure 1: Mean ice breakup date over ten-year periods from 1871 to 2010, where shading represents the number of days after the**
**31st of August when ice breakup occurs. The blue, purple, and red contour lines respectively represent a breakup date by the first**
**of June, the first of May, and the first of April. Black dots indicate the observation sites for each ten-year period.**



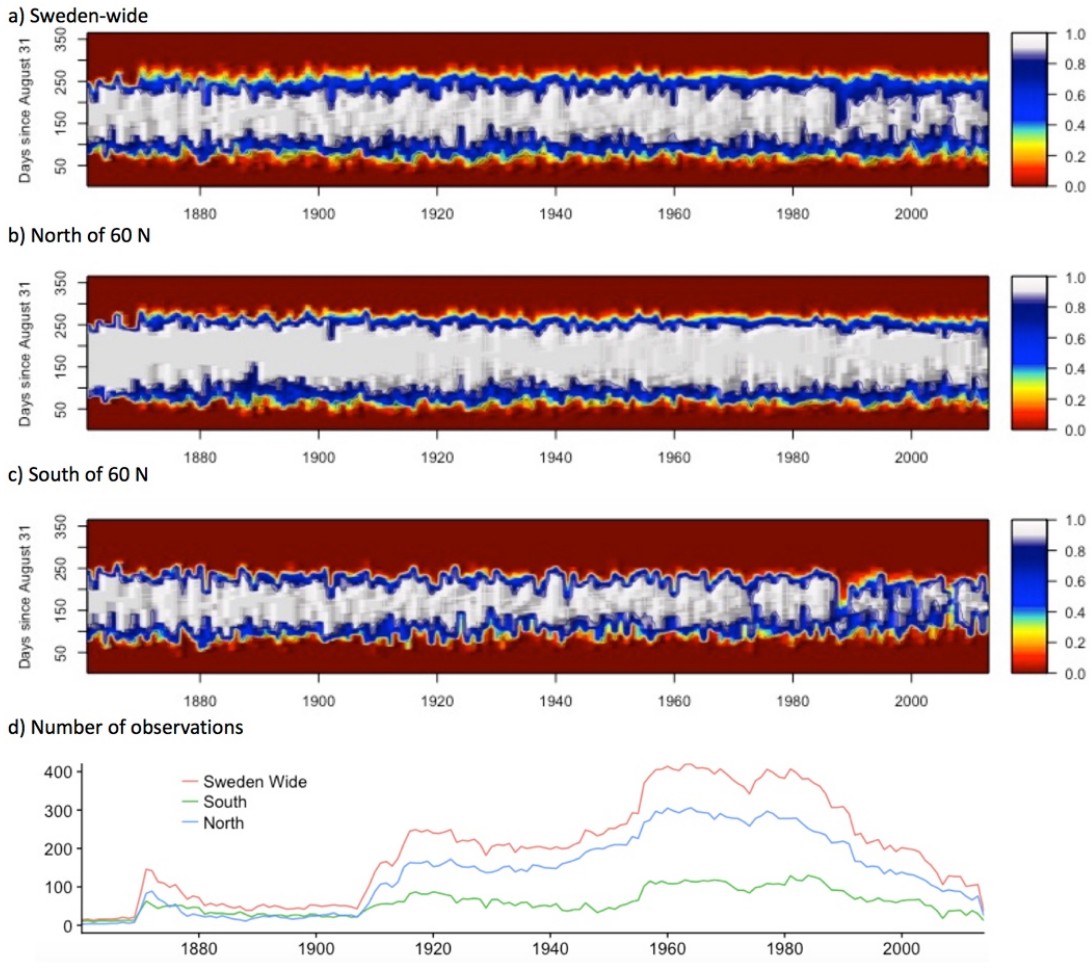

**Figure 2: Fraction of the total number of observed lakes and rivers with ice cover each day of the ice year (September-August) for all observed water bodies from a) the entirety of Sweden and b) north and c) south of 60°N in Sweden. White-grey colouring represents days with 90% - 100% (i.e., 0.9-1.0) of lake-ice area covered. Plot d) displays the number of observations each year.**





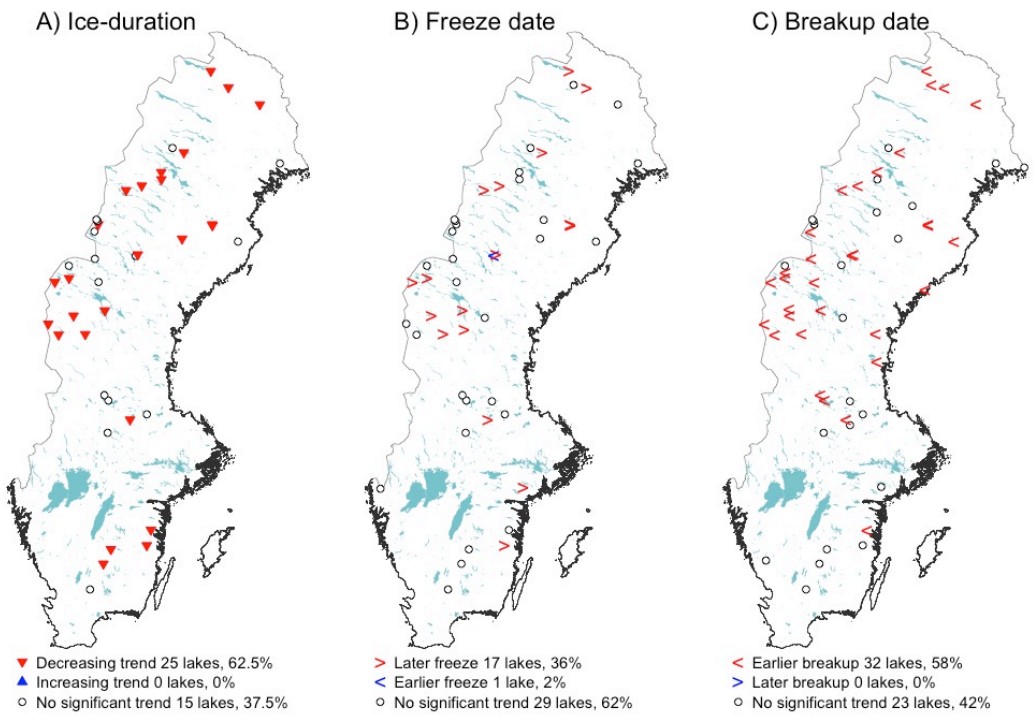


**Figure 3: Trends in a) ice duration, b) freeze date, and c) breakup date during 1913-2014. The number of water bodies and the corresponding percentages of observation locations with statistically significant (p<0.05) or insignificant trends are indicated.**



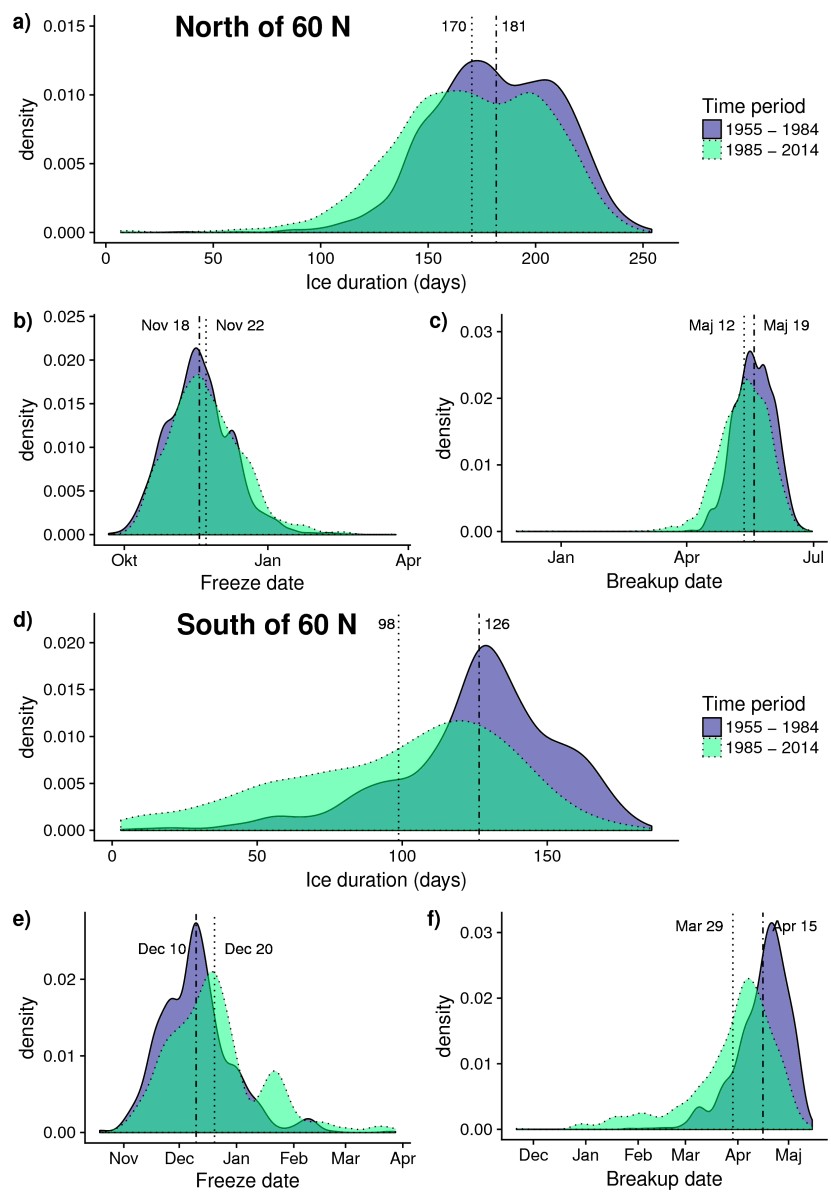

**Figure 4: Probability density functions for ice duration (a,d) and freeze (b,e) and breakup dates (c,f) from northern (a-c) and southern (d-f) Sweden during 1955-1984 and 1985-2014. Vertical lines represent mean values, where dot-dashed and dotted lines represent the former and latter 30-year periods, respectively.**





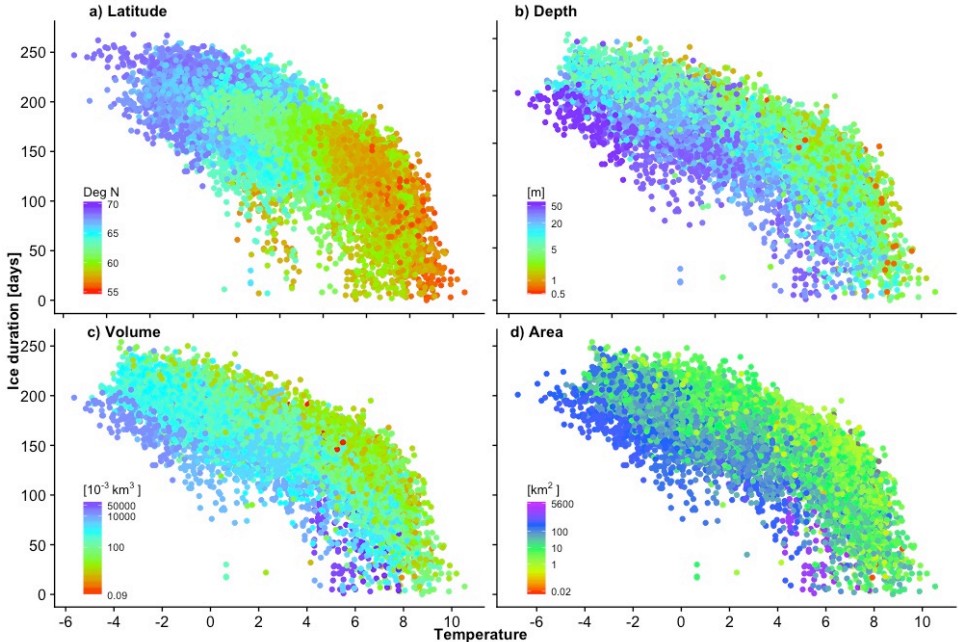


**Figure 5: Ice duration (days) versus mean annual air temperature (September-August) (in degrees Celsius). Colours indicate the water body a) latitude, b) mean depth, c) volume, and d) area.**



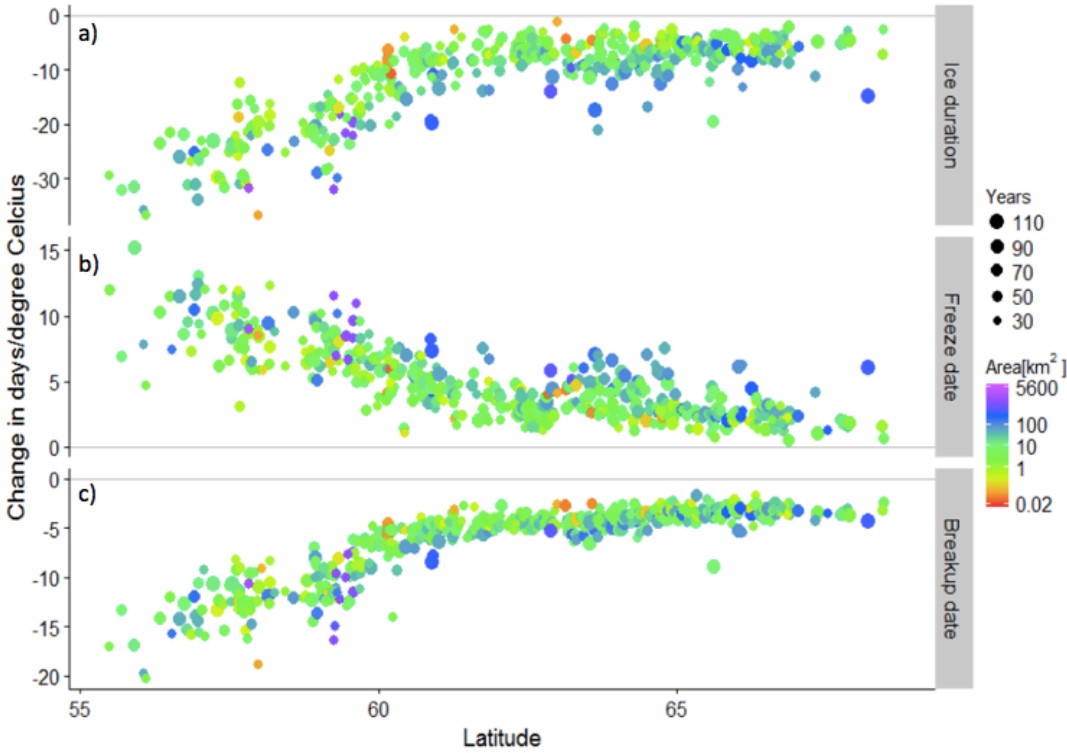

**Figure 6: Change (in days per 1°C increase in air temperature) in a) ice duration relative to annual mean temperature, b) freeze date relative to mean temperature from October-December, and c) breakup date relative to mean temperature from March-May based on the linear regression of air temperature and observation date each year for each lake. For the freeze and breakup, negative (positive) values represent earlier (later) dates with increasing temperature. Marker colours represent the area of the lake, while marker sizes denote the number of years with both lake ice and temperature observations.**