# Peer review of "Warming Climate Shortens Ice Durations and Alters Freeze and Breakup Patterns in Swedish Water Bodies"

_The Cryosphere, 2021_

## Referee Comment (RC2)

**Warming Climate Shortens Ice Durations and Alters Freeze and Breakup Patterns in Swedish Water Bodies**

**Authors**: Sofia Hallerbäck, Laurie S. Huning, Charlotte Love, Magnus Persson, Katarina Stensen, David Gustafsson, and Amir AghaKouchak

**Journal:** *The Cryosphere*

1.      **Summery:** In this paper Hallerbäck et al investigated icing pattern in different lakes and rivers in Sweden. By application of MK-trend test, they analyzed 1)ice duration, shifts in 2)freezing and 3)breakup time. They concluded that the mean observed ice durations is decreased considerably in northern (11 days) and southern Sweden (28 days). Additionally they showed while freezing is occurring later in the year, break up time happens sooner.

**2. General Comment**

I find this very paper interesting. The presented data is indeed unique. Given the spatial coverage of Sweden, this paper cast light on a very important process (icing) for climate change impact assessment. The data is well presented. To make the paper even stronger and provide with a rigid scientific backbone I have following remarks: 1- there are a few points where I think providing with more discussion, clarification and information is necessary. This specifically is the case for part 3-2 and 3-3. 2- Some sentence needs to be brushed up. Specifically, in introduction part, the references should undergo cosmetic adjustments 3- The time period for different analysis are varying through study. It is worth justification (or reminding) the readers why specific periods are chosen for each analysis.

 Considering the extent of my comments and the capacity of this work, I suggest **minor revision**.

**3. Specific Comment**

**Abstract**: The abstract is generally clear and is in line with the content of the paper.

L16: please change the order of ice duration, later freeze and earlier break up to 'earlier break up, shorter icing duration and later freeze' to match the order of your result section.

L17: What do you mean by dates from 1913-2014? Please clarify. Also, please specify to what periods are these changes relative to.

L19: Maybe it is not necessary to have this sentence: the rate of the change is … . From the previous sentence, one can easily deduce the ratio of change.

L21: please specify by exact dates when was 'previously observed' period.

L23: What do you mean by 1 degree increase? Do you mean from now (2021) onwards if we have 1 degree of more increase in temperature (in what time span would this increase happened?), then we would see decrease in ice duration by the end of 2100? Please clarify.

**Introduction:** Please check all references to be inline with one of the referencing styles. In line 27 ( and some others) for example, the references are mentioned separately.

L 76- 87: These two paragraphs would fit better in method and data part in my opinion. Either merge this part to the existing content in the method part or consider removing it.

Then in the last paragraph maybe it is of use again to specify why your study is different from Weyhenmeyer et al 2004. Remind of all the key things that you additionally looked at (and maybe they did not).

**Methodology**:

L90-91: Is Thorne river, the only data provided by SYKE? Then consider rephrasing the first two lines to: 'the refreeze/ice data records are provided from two sources: Most data consisting of river and lake ice record is provided by SMHI while Thorne river data is obtained from SYKE'. If more data is provided by SYKE, consider adding number/percentages.

L92: consisting of [number of] lakes and [number of] river.

L94: Given the data spans 1700-2004 please specify why the data from 1860-2014 is presented in figure 3.

L95: Consider rephrasing please. Suggestion: ' systematic observation of lake icing started in 1870 by and observer responsible for manual monitoring a specific lake …'

L99: not necessary to start a new paragraph.

L103: This sentence is unclear to me. Please consider simplifying it. Suggestion: 'The ice year is defined from $1^{st}$ of September to $31^{st}$ of August.' If this is what you intended to say in the first sentence.

L107-L111: In my opinion this part needs some shuffling of text. Uncertainty and the issue with numerous observation  has already been mentioned previously in this part of the text. Please consider either moving them to where you already talked about these points or removing them.

**Statistical tests, trends and spatial analysis**

For the sake of part 3.3 (changes in the timing of ice cover), it might be of use to add some test to check if data are coming from the same distribution with tests such as chi test or Mann Whitney u test. Please consider adding such tests for reasons I will explain in the relevant part.

**Results:**

**3.1** – I personally find figure 1 vey interesting. I also think that a bit explanation of the results would be beneficial. For example it seems like there were fewer observation sites in the first 4 decades. This might affect blue, purple and red contour lines vicinity and the area covered by contours (please note that in figure representing 1881-1890 period, the blue line does not intersect with Sweden boundaries). It is beneficial to remind readers of the changed number of observational points and its potential effect and uncertainty on this presentation.

L134:L 145: since quite a big portion of result is this section is allocated to extracted data from Torne river and Väserås Fjäld, I think it is beneficial to add a map with the location of these two waterways for readers who are not familiar with exact Geography of Sweden. Please consider adding such map.

L:138: Either try to add more references other than Sharma 2016 here to back up such findings or consider moving this part to discussion where you would thoroughly put your finding into context.

**3.2:** L157. Please consider rephrasing the first sentence of the paragraph to a more formal language.

L158: What is ice phenology? Please consider bringing this technical term either earlier in the manuscript (in the introduction where you already have it) or remove it because 'breakup dates' has already been used numerous times.

Please also specify why this particular period is used (1913-2014).

Figure3: **This is one of my main comments.** I personally have some difficulties understanding why these particular points are selected**.** First I thought in total, each map contains 40 water bodies that represent some significant behavior in either of the three characteristics.

But I the data does not add up to 40 in all three (for example please check figure 3 which seems to have 55 water bodies according to notation under the map). Additionally, there are some points that do not add show up in all maps. For example, note the data point with insignificant trend in the west coast of Sweden (Maybe around Trollhätten) in figure B. I could not find this point in any of the other maps.

Can you either revisit the maps or make a more clear explanation of what these maps show?

L170: The sentence is unclear. Please rephrase.

L174:  It is very difficult to find blue mark in figure 3.B. Consider reshaping or using another marker.

L175: Can you add reference to SMHI's temperature data? Also please make a connection between this statement and previous sentence to explain how this is relevant for this study.

L180: What is the main finding of these studies? And how does it help to put current paper into context for a broad audience?

**3.3: This is one of my main comments.** Although the change in mean (first moment), most probably is significant, it is aslo important to back up such finding by analyzing if the changes in ice duration is significantly different in two samples. For example you can use Z test, Mann–Whitney $U$ test or Wilcoxon rank-sum test. Without such analysis, it is not easy to know if this changes are happening randomly or there is more to it.

L186: please mention that these two periods are specifically selected to cover 30 years climatic normal periods otherwise it is not quite clear why these periods are selected.

L198: I am not sure what does this part means: 'resulting from an increase (of what?) in extremely short duration'

L203-204: I am not entirely sure why do we have this sentence here : 'Nonetheless, other lake characteristics such as mean depth, volume, and area also influence the duration of ice cover and its freeze and breakup dates'. Please clarify or make connection to other parts of the study.

3.4 L 212: I am not entirely sure why do you have 30,000 observations. is it derived from number from timeseries of 464 lakes? If it is the case isn't it more straight forward if the start and end point of observed temperature was reported?

L215: I am not sure what part of appendix you refer to.

Figure 5:**This is also another major comment I have:** I personally find it easier to understand these plots if different colormaps were used for each plot. From these 4 plots I cannot really see the effect from Depth and volume. In fact it still seems like the pattern is highly moderated by latitude compared to Depth or Volume. I think the connection/disconnection between area,

volume and depth to ice duration needs a little bit more discussion or better clarification/ justification (if there is any).

In fact it is also quite important for the readers to know if Swedish lakes are being moderated by their characteristics or are only controlled by climatic patterns or location.

L222: same as what I already commented in abstract part, in what period would 1 degree increase in temperature happen? Please clarify. Generally I am not sure how these ratio is being captured and what has happened here? Did you extrapolate data to future? Please add more detail to the content here.

According to Figure 6, does it mean that e.g., in ice duration some catchments encounter -30 day ice duration in case of 1 degree (per what period?) temperature rise. If this is the case please add more information about what this plot actually represents.

L251. 252: how does the relationship between breakup date and mean temperature representation via arc cosine function relate to your study? Please clarify.

L260: please clarify what do you mean by smaller temperature amplitude.

Sincerely

**Faranak Tootoonchi**

---

## Author Comment (AC2)

Thank you for your thoughtful comments and suggestions. We believe that your input has improved the manuscript. Below you can find our point-by-point respond to your comments and suggestions.

**1. Summery:** In this paper Hallerbäck et al investigated icing pattern in different lakes and rivers in Sweden. By application of MK-trend test, they analyzed 1)ice duration, shifts in 2)freezing and 3)breakup time. They concluded that the mean observed ice durations is decreased considerably in northern (11 days) and southern Sweden (28 days). Additionally they showed while freezing is occurring later in the year, break up time happens sooner.

**2. General Comment**

I find this very paper interesting. The presented data is indeed unique. Given the spatial coverage of Sweden, this paper cast light on a very important process (icing) for climate change impact assessment. The data is well presented. To make the paper even stronger and provide with a rigid scientific backbone I have following remarks:

Response: Thank you for the positive feedback regarding our paper and acknowledging the unique aspects of our work.

1- there are a few points where I think providing with more discussion, clarification and information is necessary. This specifically is the case for part 3-2 and 3-3.

Response:  We agree that more discussion, clarification and information would be beneficial to increase the transparency and understanding of the reader. Answering your specific comments below, we attempt to clearly describe 3-2 and 3-3. The suggested changes will be added to the revised version.

2- Some sentence needs to be brushed up. Specifically, in introduction part, the references should undergo cosmetic adjustments.

Response: Thank you for addressing this and the changes will indeed improve the manuscript.

3- The time period for different analysis are varying through study. It is worth justification (or reminding) the readers why specific periods are chosen for each analysis.

Response: We agree with the reviewer and we acknowledge that justifying and reminding the reader of the different time periods will improve the comprehension of the paper.

The main reason for using different time periods is that the lengths of records vary significantly. Selecting a period of overlap leads to elimination of significant data points, some dating back to the 18th century. We designed the periods such that we can incorporate as much of the data as possible in our analysis. The main question was: do we see a shift in timing of ice cover including freeze and breakup observations? To answer this question, we looked at longer time periods using the water bodies with up to 100 years freeze and breakup data, see Figure 3. For analysis of trends in each lake we used all available data, resulting in Figure 1. In Figure 4, we wanted to look at the last 30 years and determine if we could identify a difference from the previous 30-year period. In the revised version, we make sure that the time periods are clearly defined in each caption and also within the manuscript.

Considering the extent of my comments and the capacity of this work, I suggest **minor revision**.

**3. Specific Comment**

**Abstract**: The abstract is generally clear and is in line with the content of the paper.

L16: please change the order of ice duration, later freeze and earlier break up to 'earlier break up shorter icing duration and later freeze' to match the order of your result section.

Response: Thank you; we will address this comment in the revised version. Also see answer to L17 for clarification.

L17: What do you mean by dates from 1913-2014? Please clarify. Also, please specify to what periods are these changes relative to.

Response: We agree that the sentence in the abstract was not clear and will clarify it in our revision. We referred to the results of trend analysis using freeze and breakup date observations from 1913 to 2014.

L19: Maybe it is not necessary to have this sentence: the rate of the change is ... . From the previous sentence, one can easily deduce the ratio of change.

Response: Good point. We will remove this in the revised version.

L21: please specify by exact dates when was 'previously observed' period.

Thank you, we agree that clarification is needed. We refer to Figure 4 and the period 1955-1984 as the 'previously observed' period. We will clarify this in the revised version.

L23: What do you mean by 1 degree increase? Do you mean from now (2021) onwards if we have 1 degree of more increase in temperature (in what time span would this increase happened?), then we would see decrease in ice duration by the end of 2100? Please clarify.

Response: Here, we looked at historical observations which include several degrees of variability. Building the relationship between the temperature and ice patterns (Figure 6) ,we can quantify the effect of 1 C change in the historical period. This approach allows us to relatively compare different regions. As seen in Figure 6, water bodies in southern

Sweden show a much faster shift in ice pattern with each degree warming climate, compared to the northern part. We did not use climate model simulations here and the results should not be used to predict the future (e.g., 2100). However, our results highlight the sensitivity of ice properties to the historical change in observations.

**Introduction:** Please check all references to be inline with one of the referencing styles. In line 27 ( and some others) for example, the references are mentioned separately.

Response: Thank you for the comment.We agree that it should be addressed in the revision.

L 76- 87: These two paragraphs would fit better in method and data part in my opinion. Either merge this part to the existing content in the method part or consider removing it.

Response: Thank you for a great comment. We agree that the paragraphs would fit better in the Method and Data section and will change accordingly.

Then in the last paragraph maybe it is of use again to specify why your study is different from Weyhenmeyer et al 2004. Remind of all the key things that you additionally looked at (and maybe they did not).

Response: We agree and we will address this issue in the revised version. Briefly, we have included more recent observations, offered a more comprehensive trend assessment and also quantified the sensitivity of ice properties to a unit degree temperature change.

**Methodology**:

L90-91: Is Thorne river, the only data provided by SYKE? Then consider rephrasing the first two lines to: 'the refreeze/ice data records are provided from two sources: Most data consisting of river and lake ice record is provided by SMHI while Thorne river data is obtained from SYKE'. If more data is provided by SYKE, consider adding number/percentages.

Response: The data was provided to us by SMHI, however the original data from Torne river stems from SYKE.

L92: consisting of [number of] lakes and [number of] river.

Response: We will add the suggested information in the revised version.

L94: Given the data spans 1700-2004 please specify why the data from 1860-2014 is presented in figure 3.

Response: Please note that  the data from 1700-1860 is only limited to a few water bodies. The below figures (Figures R1-R2) show the lengths of available observations.

[Figure]

Figure R1. Number of observations of breakup (green), freeze (blue), and both freeze and breakup (i.e., duration) for the same year and water body (red) in Sweden.

[Figure]

Response: Figure R2. Number of observations of breakup (green), freeze (blue), and both freeze and breakup (i.e., duration) for the same year and water body (red), from 1700 to 1860.

L95: Consider rephrasing please. Suggestion: ' systematic observation of lake icing started in 1870 by and observer responsible for manual monitoring a specific lake ...'

Response: Thank you; we agree.

L99: not necessary to start a new paragraph.

Response: Thank you; we agree.

L103: This sentence is unclear to me. Please consider simplifying it. Suggestion: 'The ice year is defined from 1st of September to 31st of August.' If this is what you intended to say in the first sentence.

Response: Thank you, that is what we mean and the suggested clarification is good.

L107-L111: In my opinion this part needs some shuffling of text. Uncertainty and the issue with numerous observation has already been mentioned previously in this part of the text. Please consider either moving them to where you already talked about these points or removing them.

Response: We agree with this suggestion and we will address it in the revised version.

**Statistical tests, trends and spatial analysis**

For the sake of part 3.3 (changes in the timing of ice cover), it might be of use to add some test to check if data are coming from the same distribution with tests such as chi test or Mann Whitney u test. Please consider adding such tests for reasons I will explain in the relevant part.

Response: We agree. We will address this comment in the revised version (see also our response to 3.3 below).

**Results:**

**3.1** – I personally find figure 1 very interesting. I also think that a bit explanation of the results would be beneficial. For example it seems like there were fewer observation sites in the first 4 decades. This might affect blue, purple and red contour lines vicinity and the area covered by contours (please note that in figure representing 1881-1890 period, the blue line does not intersect with Sweden boundaries). It is beneficial to remind readers of the changed number of observational points and its potential effect and uncertainty on this presentation.

Response: We are glad to hear that you found the figure interesting. We will address this issue in the revised version.

L134:L 145: since quite a big portion of result is this section is allocated to extracted data from Torne river and Väserås Fjäld, I think it is beneficial to add a map with the location of these two waterways for readers who are not familiar with exact Geography of Sweden. Please consider adding such map.

Response: Thank you. We will add the suggested map in the Supplementary Materials.

L:138: Either try to add more references other than Sharma 2016 here to back up such findings or consider moving this part to discussion where you would thoroughly put your finding into context.

Response: We will address this comment in the revised version.

**3.2:** L157. Please consider rephrasing the first sentence of the paragraph to a more formal language.

Response: We agree; thank you. The first two sentences revised as: *The trend over the last century (1913-2014) was analysed using the Mann-Kendall trend test (Kendall 1938) at a 0.05 significance level. Lakes and rivers with a maximum of 20% missing observations were included in trend test analysis of ice duration (40 water bodies), freeze date (47 water bodies) and breakup date (57 water bodies).*

L158: What is ice phenology? Please consider bringing this technical term either earlier in the manuscript (in the introduction where you already have it) or remove it because 'breakup dates' has already been used numerous times.

Response: With ice phenology we refer to variability and trends in lake ice dynamics (freeze and breakup dates and the ice duration). The term is used in many other lake and river ice studies, such as Latifovic et al. (2007), Benson et al. (2011) and Jensen et al (2007). We will add a clarification of the terms earlier in the revised version.

Please also specify why this particular period is used (1913-2014).

Response: The earlier discussion about lengths of records will be added to the revised version.

Figure3: **This is one of my main comments.** I personally have some difficulties understanding why these particular points are selected. First I thought in total, each map contains 40 water bodies that represent some significant behavior in either of the three characteristics.

But the data does not add up to 40 in all three (for example please check figure 3 which seems to have 55 water bodies according to notation under the map). Additionally, there are some points that do not add show up in all maps. For example, note the data point with insignificant trend in the west coast of Sweden (Maybe around Trollhätten) in figure B. I could not find this point in any of the other maps.

Can you either revisit the maps or make a more clear explanation of what these maps show?

Response: In the revised version, we will clarify the fact that not all of the lakes have the same three duration (A), freeze (B) and breakup (3) data in each year, see answer L57. In other words, there might be breakup date observations for some years, but there may be a lack of freeze date observations during those same years (especially in the earlier years of the record). We have included the lakes with no more than 20% missing data.

L170: The sentence is unclear. Please rephrase.

Response: The sentence was "The sites with longer records more commonly exhibited a trend than those with more missing data". We agree that the sentence is not clear and should be rephrased. We included the lakes with no more than 20% missing data. Our results showed the lakes that had a more complete record (small % missing data) exhibited more significant trends.

L174: It is very difficult to find blue mark in figure 3.B. Consider reshaping or using another marker.

Response: Thank you for the comment. We will revisit the blue markers.

 L175: Can you add reference to SMHI's temperature data?

Response: We agree that a reference to the data would be beneficial. The data source is SMHI and method of harmonization is described in following papers:

Alexandersson, H. and Moberg, A. (1997), Homogenization of Swedish Temperature data. PART I: Homogeneity Test for Linear Trends. International Journal of Climatology, 17: 25–34. doi: 10.1002/(SICI)1097-0088(199701)17:1<25::AID-JOC103>3.0.CO;2-J

Moberg, A. and Alexandersson, H. (1997), Homogenization of Swedish Temperature data. PART II: Homogenized Gridded Air Temperature Compared with a Subset of Global Gridded Air Temperature since 1861. International Journal of Climatology, 17: 35–54. doi: 10.1002/(SICI)1097-0088(199701)17:1<35::AID-JOC104>3.0.CO;2-F

Moberg, A. and Bergström, H. (1997), Homogenization of Swedish temperature data. Part III: the long temperature records from Uppsala and Stockholm. International Journal of Climatology, 17: 667–699. doi: 10.1002/(SICI)1097-0088(19970615)17:7<667::AID-JOC115>3.0.CO;2-J

Also please make a connection between this statement and previous sentence to explain how this is relevant for this study.

Response: We wanted to provide context of how the temperature has increased overall in Sweden. However we agree that the sentence is a little out of context. We will move this part to the discussion section.

L180: What is the main finding of these studies? And how does it help to put current paper into context for a broad audience?

Response: We included a table in Supplementary Materials that summarizes the findings of these papers (see Table S1). Magnusson et al. (2000) look at long term trends from 1846 to 1995 for lake and river ice in the northern hemisphere. Over that 150 year period, they see a later freeze (5.8 days per 100 years) and earlier breakup (6.5 days per 100 years). Our paper updated these numbers with more recent observations.

Benson et al. (2012) analyzed trends, frequency of extreme events and variability lake-ice phenology in 75 lakes in the northern hemisphere (not specific to Sweden),

Takács (2011) and Lonita et al. (2018)  analyzed trends in ice phenology for the river Danaube.

Latifovic et al. (2007) uses remote sensing data combined with in situ data to analyse trends in lake ice phenology in Canada. From 1950-2004 they found an earlier breakup of about 0.18 days per year and later freeze of about 0.12 days per year. When they looked at the last 20 years the rate of change was on average 0.99 days per year earlier breakup and 0.76 days per year later freeze.

Jensen et al. (2007) analyzed freeze and breakup dates and ice durations in the Great Lake region of the United States during 1975-2004. They found an average change of 3.3 days per decade later freeze and 22.1 days per decade earlier breakup.

Hodgkins (2013) analyzed trends in breakup dates in New England, USA, over several different time periods 25, 50, 75, 100, 125, 150 and 175 year periods ending in 2008. Breakup dates over the last 50 years changed 1.8 days per decade.

**3.3: This is one of my main comments.** Although the change in mean (first moment), most probably is significant, it is also important to back up such finding by analyzing if the changes in ice duration is significantly different in two samples. For example you can use Z test, Mann– Whitney *U* test or Wilcoxon rank-sum test. Without such analysis, it is not easy to know if this changes are happening randomly or there is more to it.

Response: We agree with the reviewer. This information will be added in the revised version.

L186: please mention that these two periods are specifically selected to cover 30 years climatic normal periods otherwise it is not quite clear why these periods are selected.

Response: We agree. The periods will be clarified in all captions and figure descriptions.

L198: I am not sure what does this part means: 'resulting from an increase (of what?) in extremely short duration'

Response: Revised as: *Changes in timing of ice cover have in recent years shortened the mean ice duration and increased occurrence of years with a very short ice duration.*

L203-204: I am not entirely sure why do we have this sentence here : 'Nonetheless, other lake characteristics such as mean depth, volume, and area also influence the duration of ice cover and its freeze and breakup dates'. Please clarify or make connection to other parts of the study.

Response: We should clarify. We wanted to remind the reader that there is uncertainty in the results stemming from missing data. The missing data could result in different water bodies being somewhat unequally represented in the two different time periods, or samples. Further, the location and other physical characteristics can influence the results.

3.4 L 212: I am not entirely sure why do you have 30,000 observations. is it derived from number from timeseries of 464 lakes? If it is the case isn't it more straight forward if the start and end point of observed temperature was reported?

Response: We should clarify. The sentence L 211-L 215 is revised as: Figure 5 characterizes the relationship between the local mean annual air temperatures (September - August) and ice duration. Air temperature observations are derived from the Climate Research Unit (CRU) v 3.23 (Harris et al. 2014).

L215: I am not sure what part of appendix you refer to.

We will address this comment in the revised version. In the supplementary materials (Figure S3) the relationship with different time periods of temperature and latitude is presented.

Figure 5: **This is also another major comment I have:** I personally find it easier to understand these plots if different colormaps were used for each plot. From these 4 plots I cannot really see the effect from Depth and volume. In fact it still seems like the pattern is highly moderated by latitude compared to Depth or Volume. I think the connection/disconnection between area, volume and depth to ice duration needs a little

bit more discussion or better clarification/ justification (if there is any). In fact it is also quite important for the readers to know if Swedish lakes are being moderated by their characteristics or are only controlled by climatic patterns or location.

Response: We agree that it would help to clarify the connection with latitude, depth, volume and area respectively. In Sweden, the climate is much colder in the northern regions compared to the southern. The most northern lakes are frozen for a longer period than southern Sweden lakes. In fact, some southern lakes might not freeze at all in the winter. This results in latitude being the overall dominant factor compared to the size of the lake.

L222: same as what I already commented in abstract part, in what period would 1 degree increase in temperature happen? Please clarify. Generally I am not sure how these ratio is being captured and what has happened here? Did you extrapolate data to future? Please add more detail to the content here.

Response: See answer to L23.

According to Figure 6, does it mean that e.g., in ice duration some catchments encounter -30 day ice duration in case of 1 degree (per what period?) temperature rise. If this is the case please add more information about what this plot actually represents.

Response: See answer to L23. To clarify further: Figure 6 plots the ratio of change historically observed per degree e.g. in for some lakes in southern Sweden a degree temperature increase historically decreases the ice duration by approximately 30 days.

L251. 252: how does the relationship between breakup date and mean temperature representation via arc cosine function relate to your study? Please clarify.

Response: We did not analyze that relationship.

L260: please clarify what do you mean by smaller temperature amplitude.

Response: With the phrase, we refer to the temperature amplitude as the peak-to-peak difference in temperature for a specific site. Smaller temperature differences indicate that the  site does not experience extreme shifts of temperatures over the year.